



# Utilizing an Electrical Low Pressure Impactor to Indirectly Probe Water Uptake via Particle Bounce Measurements

Kevin B. Fischer[1], Giuseppe A. Petrucci[1]

[1]Department of Chemistry, The University of Vermont, Burlington, VT, 05405, USA

*Correspondence to*: Giuseppe A. Petrucci (giuseppe.petrucci@uvm.edu)

**Abstract.** Secondary organic aerosol (SOA), formed through oxidation of volatile organic compounds (VOCs), display complex viscosity and phase behaviors influenced by temperature, relative humidity (RH), and chemical composition. Here, the efficacy of a multi-stage electrical low pressure impactor (ELPI) for indirect water uptake measurements was studied for ammonium sulfate (AS) aerosol, sucrose aerosol, and α-pinene derived SOA. All three aerosol systems were subjected to

greater than 90% chamber relative humidity, with subsequent analysis indicating persistence of particle bounce for sucrose aerosol of 70 nm (initial dry diameter) and α-pinene derived SOA of number geometric mean diameters between 39 nm and 136 nm (initial dry diameter). On the other hand, sucrose aerosol of 190 nm (initial dry diameter) and AS aerosol down to 70 nm (initial dry diameter) exhibited no particle bounce at elevated RH. Partial drying of aerosol within the lower diameter ELPI impaction stages, where inherent and significant RH reductions occur, is proposed as one explanation for particle bounce

persistence.

## 1 Introduction

Secondary organic aerosol (SOA) is formed via oxidation of volatile organic compounds (VOCs) to produce semi-, low- and very low-volatility products that can nucleate to form new particles or partition/condense into/onto existing particles, adding to the overall aerosol mass burden (Hallquist et al., 2009; Jimenez et al., 2009). SOA constitutes a significant portion of ambient

fine aerosol and accounts for a large fraction of the organic aerosol burden, which itself makes up a sizeable portion (20 – 90%) of submicron particulate mass (Kourtchev et al., 2016; Jimenez et al., 2009; Kanakidou et al., 2005; Hallquist et al., 2009). Understanding the viscosity of atmospheric SOA is important to improving the predictive capabilities of their atmospheric impacts (Koop et al., 2011; Bateman et al., 2016; Shiraiwa et al., 2017; Li et al., 2020). Water vapor, a key and ubiquitous component of the troposphere (Sherwood et al., 2010), can act as a plasticizer when taken up by SOA, subsequently

lowering particle viscosity (Koop et al., 2011; Zobrist et al., 2008). SOA water uptake in the air is largely dependent on the particle hygroscopicity (Jimenez et al., 2009; Tang et al., 2019; Chu et al., 2014), which impacts the SOA's aerodynamic and optical properties as well as heterogeneous chemical reactions (with gas phase reactants), and cloud droplet activation (cloud condensation nuclei activity) (Cappa et al., 2011; Michel Flores et al., 2012; Michaud et al., 2009; Heintzenberg et al., 2001; Chu et al., 2014; Wu et al., 2020; Zhao et al., 2016; Burgos et al., 2019). RH over 90% is common, especially in the vicinity



of clouds (Hamed et al., 2011) and recent work has expanded the focus to uptake of water by SOA under sub-saturation levels, where relative humidity (RH) is below 100%, and the resulting changes in viscosity (Pajunoja et al., 2016; Berkemeier et al., 2014; Reid et al., 2018). Uptake of enough water can solubilize a particle if RH becomes high enough, facilitating its response to changes in gas-phase composition. Thus, RH, along with chemical composition and temperature, has been deemed a controlling factor of aerosol phase state (Koop et al., 2011; Krieger et al., 2012; Martin, 2000; Reid et al., 2018).


While several methods for investigating particle hygroscopicity and viscosity under sub-saturated conditions have been reported (see Tang et al. (2019) for a recent review on the topic), it is difficult to extrapolate from these methods to understand how RH might influence the reactivity of chemically complex SOA at atmospherically relevant mass loadings, particle sizes and timescales. Often, existing methods require large particles (several μm), unrealistically high SOA mass loading ($C_{SOA}$) in

the hundreds to thousands of μg m$^{-3}$, specific particle size selection via a differential mobility analyzer (therefore inherently excluding much of the SOA population), and aerosol collection times of hours to days (Grayson et al., 2016; Renbaum-Wolff et al., 2013; You et al., 2012; Tang et al., 2019; Bateman et al., 2014; Bateman et al., 2015). For example, Grayson et al. (2016) used the poke-and-flow technique to measure the viscosity of α-pinene derived SOA at $C_{SOA}$ between 120 and 14,000 μg m$^{-3}$. Maclean et al. (2021) recently completed a similar poke-and-flow study with β-caryophyllene derived SOA. Kidd et al. (2014)

found similar results by studying SOA impaction patterns onto a germanium crystal, which also permitted FTIR spectroscopic analysis of chemical composition. Furthermore, particle sizes studied were limited to tens of micrometers by the measurement methods. Jarvinen et al. (2016) reported on the use of an *in situ*, real-time optical method to detect the viscous state of α-pinene SOA particles and measured their transition from the amorphous highly viscous state to states of lower viscosity. Their method is based on the depolarising properties of laboratory-produced non-spherical SOA particles and their transformation to non-

depolarising spherical particles at RH values near the deliquescence point (where sufficient water is absorbed to form an effectively aqueous solution). Particles of at least 600 nm geometric mean diameter (GMD) were needed to obtain measurable scatter signals. From reported parameters for a typical number size distribution of their SOA (GMD = 600 nm, geometric standard deviation, GSD = 1.15 and total number density, N = 10,000 cm$^{-3}$), we can estimate a total $C_{SOA}$ of 1,570 μg m$^{-3}$.

Thus, there is a need to estimate particle viscosity for SOA subjected to different RH that is measured in situ, in real-time, and at atmospherically relevant mass loadings and particles sizes. Our previously developed method of inferring particle viscosity through quantification of particle bounce from an impaction surface using an electrical low pressure impactor (ELPI+) (Jain and Petrucci, 2015) is in principle analogous to similar methods developed previously (Bateman et al., 2014; Saukko et al., 2012a; Virtanen et al., 2011). However, the ELPI method differs significantly in its application and output as it takes advantage

of the smooth and sintered impaction surface characteristics to calculate bounce factor (BF) with high temporal resolution to infer viscosity changes in a continuously evolving SOA population, while simultaneously eliminating the need for additional instrumentation to independently measure size distributions (Jain and Petrucci, 2015).



Here, our ELPI setup was utilized to probe particle bounce for aerosol that was subjected to high RH (> 90%) conditions to
determine the viability of indirect measurements of water uptake. Ammonium sulfate (AS) aerosol, sucrose aerosol, and SOA
derived from the ozonolysis of α-pinene were injected or generated in a batch type 775L Teflon reaction chamber fitted with
a humidifier and RH sensor. Particle bounce data was collected and compared with previous literature reports to probe the
efficacy of the ELPI for the measurement of humidified particles.

## 2 Materials and Methods

All experiments were conducted in a 775 L Teflon batch type reaction chamber under ambient temperature and atmospheric
pressure. For chamber humidification, dry, particle free air was humidified by passage through two, 4-L glass containers
containing deionized water. The first container was maintained at 35 °C water temperature via a hot plate, while the second
(immediately before injection into the chamber) was held at room temperature. RH inside the chamber was monitored using a
humidity sensor (HMT 130, Vaisala Corp., Helsinki, Finland). Prior to all experiments, dry, zero air was used for chamber
flushing (after $H_2O_2$ passivation with UV lamps) until background aerosol mass and number concentrations were well below
0.01 µg m$^{-3}$ and 10 particles cm$^{-3}$, respectively. Particle mass and number concentrations as well as size distributions were
measured using a scanning mobility particle sizer (SMPS 3082, TSI Inc., Shoreview, MN, USA) operating with sheath and
aerosol flows of 4.0 and 0.4 L min$^{-1}$, respectively. For α-pinene derived SOA experiments, SMPS flows of 10 and 1.0 L min$^{-1}$, respectively, were used.

### 2.1 Ammonium Sulfate, Sucrose Aerosol Deliquescence and Efflorescence

Ammonium sulfate (AS) and sucrose were purchased from Fisher Scientific and used without further purification. AS and
sucrose aerosol were produced by pneumatic nebulization of an aqueous solution (7.35% w/v) using a V-groove nebulizer
(J.E. Meinhard Associates, Santa Ana, CA, USA) and dried by passage through a diffusion drier packed with silica gel. The
RH of aerosol at the exit of the diffusion dryer was <10%. Following the diffusion dryer, the dried aerosol was injected into a
775 L Teflon chamber held at an initial RH of either <20% (for deliquescence experiments) or >90% (for efflorescence
experiments) for polydisperse populations. For monodisperse populations, size selection was carried out after the diffusion
dryer using a differential mobility analyzer (DMA), after which the monodisperse aerosol was led through an x-ray based
neutralizer and then into the chamber. For preparation of a high RH chamber or measurement of the deliquescence curve, the
RH was increased by the introduction of air saturated with water vapor (as described above) either before or following aerosol
injection, respectively. For measurement of the efflorescence curve, initial RH was >90% before aerosol injection. The RH
was then continuously decreased by an influx of dry, particle free air.

### 2.2 α-pinene Derived SOA Generation and Subsequent Humidification

α-pinene was purchased from Sigma-Aldrich and used without further purification. A glass micro-syringe was used to
quantitatively transfer α-pinene aliquots into a glass three-neck flask that was placed in a hot water bath. The liquid phase α-



pinene content within the flask was visually monitored as a flow of dry zero air carried the volatilized α-pinene into the 775 L

Teflon reaction chamber. Once all gaseous α-pinene was introduced, the dry zero air flow was shut off (typically after 10 – 20

minutes). The three-neck flask was sealed except for the dry zero air inlet and an outlet leading directly to the reaction chamber.

To initiate ozonolysis, ozone was produced with a commercial generator (1KNT, Enaly, Shanghai, China) using dry, particle

free air. Ozone was injected by diverting the output flow of the generator first through an airflow splitter to dilute the ozone

concentration, and subsequently to the chamber for a pre-determined time pulse to yield the desired ozone concentration.

Typical injection pulses were in the range of 5 – 15 seconds. A summary of the α-pinene derived SOA experiments is provided

in Table 1. RH was subsequently increased by the introduction of air saturated with water vapor (as described above).

| Experiment | α-pinene Mixing Ratio (ppb) | Ozone Mixing Ratio (ppb) | $C_{SOA}$ (μg m$^{-3}$) | GMD (nm) |
|:---:|:---:|:---:|:---:|:---:|
| A | 195 | 150 | 5 | 39 |
| B | 195 | 175 | 16 | 79 |
| C | 195 | 250 | 52 | 102 |
| D | 195 | 400 | 217 | 108 |
| E | 195 | 650 | 626 | 119 |
| F | 195 | 750 | 702 | 101 |
| G | 391 | 2,000 | 2,420 | 136 |

**Table 1.** Mixing ratios used for each α-pinene derived SOA experiment. Both SOA mass loading ($C_{SOA}$) and number geometric

mean diameter (GMD; from SMPS instrument) were observed immediately prior to initiation of humidification.

**2.3 Electrical Low Pressure Impactor**

An electrical low pressure impactor (ELPI+, Dekati, Kangasala, Finland) was operated at a constant flow rate of 10 L min$^{-1}$

and sampling was done directly from the Teflon reaction chamber. The ELPI consists of 15 stages (13 active impaction stages,

1 active filter stage) with sequentially smaller cut-off impaction diameters ($D_{50}$; particle size with a 50% collection efficiency)

ranging from 10 μm (stage 15) to 6 nm (stage 1/filter stage). Stage 15 is not measured electrically. The ELPI classifies particles

based on their aerodynamic diameter. Operation of the ELPI can be divided into three main sections: (1) unipolar charging of

the aerosols, (2) size classification of these charged aerosols in a Berner type low pressure cascade impactor, and (3) electrical

measurement of collected particles by a series of electrometers (one for each stage). Further operational and calibration details,

including corrections for diffusion and space charger losses, can be found elsewhere (Keskinen et al., 1999; Virtanen et al.,

2001; Järvinen et al., 2014; Jain and Petrucci, 2015). In the modified method utilized here, contrary to the previously published

method (see Supplement) (Jain and Petrucci, 2015), only the smooth impaction stages were utilized, which allow for particle

bounce to occur.



To qualitatively assess relative changes in aerosol viscosity, the fractional current of the filter stage, channel 1 ($\Delta i_{\text{fractional, t}}$) at

any time point (t) was determined by calculating changes in the measured current of channel 1 ($i_{\text{Ch}_{1,t}}$) as compared to the total

current summed across all channels ($\sum_{n=1}^{n=14} i_{\text{Ch}_{n,t}}$):

$$\Delta i_{\text{fractional, t}} = \frac{i_{\text{Ch}_{1,t}}}{\sum_{n=1}^{n=14} i_{\text{Ch}_{n,t}}} \quad \quad (1)$$

Here, it is assumed that all particles that bounce from any preceding stage terminate on the filter stage. As particles become

less viscous due to processes such as adsorption/absorption of water, the $\Delta i_{\text{fractional, t}}$ will decrease as a result. Note that this is

not a measure of the absolute particle bounce, which can be calibrated to provide a quantitative estimate of the viscosity, but

rather a relative change in particle bounce as a function of changing environmental conditions within the chamber. A

comparison to the bounce factor method (Jain and Petrucci, 2015) is shown in Fig. S1.


Importantly, an RH drop occurs within the lower impaction stages due to the low pressure condition needed to avoid shifts in

flow rates that would cause changes to the setpoint aerodynamic diameters. This RH drop is proportional to the pressure drop

and becomes significant at stage 7 ($D_{50} = 255$ nm) and below (Table 2). As discussed below, this impactor RH drop can affect

the bounce of those particles that possess an efflorescence relative humidity (ERH) or equivalent drying RH above the impactor

stage RH, even though the particle residence time in these low pressure impaction stages is 50.5 ms and below for each stage.








| Stage | $D_{50}$ (μm) | $t_{res}$ (ms) | $P_n$ (kPa) | $RH_n$ (%) |
|---|---|---|---|---|
| Chamber | | | 101.33 | 100 |
| 15 | 9.87 | 0 | 101.32 | 99.99 |
| 14 | 5.36 | 102.9 | 101.3 | 99.97 |
| 13 | 3.65 | 73.6 | 101.25 | 99.92 |
| 12 | 2.47 | 57.8 | 101.19 | 99.86 |
| 11 | 1.63 | 57.1 | 101.01 | 99.68 |
| 10 | 0.947 | 62.2 | 100.5 | 99.18 |
| 9 | 0.602 | 58.5 | 99.59 | 98.28 |
| 8 | 0.381 | 55.2 | 97.21 | 95.93 |
| 7 | 0.255 | 50.5 | 88.8 | 87.63 |
| 6 | 0.155 | 39.1 | 68.86 | 67.96 |
| 5 | 0.0941 | 22.2 | 38.44 | 37.94 |
| 4 | 0.0528 | 12.7 | 21.86 | 21.57 |
| 3 | 0.0296 | 5.6 | 9.73 | 9.60 |
| 2 | 0.0161 | 2.4 | 4.48 | 4.42 |
| 1 | 0.006 | 1 | 4 | 3.95 |

**Table 2.** ELPI cut point diameter ($D_{50}$; μm), residence time (ms), pressure (kPa), and calculated relative humidity (%) for each stage, assuming 100% RH and 101.33 kPa in chamber.

## 3 Results and Discussion

As the most abundant monoterpene in the troposphere, α-pinene SOA formation via ozonolysis has become an established SOA system in the atmospheric sciences community (Zhang et al., 2015a). The SOA produced under dry conditions is best described as a viscous, amorphous semi-solid with reported viscosities ranging from $10^4 - 10^9$ Pa s that can undergo moisture-induced phase transitions, given sufficiently high RH, resulting in significant viscosity changes (Zhang et al., 2015b; Grayson et al., 2016; Petters et al., 2019; Shiraiwa et al., 2011). Ammonium sulfate and sucrose have been heavily studied and are often used as surrogates for inorganic and organic aerosol components, respectively, and are both also found in the atmosphere as well (ammonium sulfate more ubiquitously than sucrose) (Ruprecht and Sigg, 1990; Theodosi et al., 2018; Froyd et al., 2019; Ott et al., 2020). Given sufficiently high RH, both engage in water uptake to undergo deliquescence (formation of an aqueous solution) (Mauer and Taylor, 2010). Conversely, water loss can occur to the point of particle drying and/or efflorescence (recrystallization at low RH) (Davis et al., 2015).





### 3.1 Ammonium Sulfate Aerosol

Efflorescence relative humidity (ERH) and deliquescence relative humidity (DRH) curves for AS aerosol (number GMD = 90 nm, initial dry diameter) were generated following typical procedures (see above). Figure 1 displays efflorescence and deliquescence behavior in terms of changes in the fractional current on channel 1, $\Delta i_{fractional, t}$ as a function of chamber RH.

As given in Eq. (1), a value closer to 0 indicates liquid like (less viscous) particles and values closer to 1 indicate solid like (more viscous) particles. Since the $D_{50}$ of channel 1 (filter stage) is 6 nm and the number GMD of the ammonium sulfate aerosol distribution was 90 nm, only particles experiencing bounce were expected to reach the filter stage (as opposed to reaching the filter stage due to their aerodynamic diameter). Therefore, values of $\Delta i_{fractional, t}$ approaching 1 indicate that all particles have reached the filter stage resulting from particle bounce from preceding stages (suggesting solid particles), while

values of $\Delta i_{fractional, t}$ approaching 0 suggest that all particles adhered to their intended impaction stages based on their aerodynamic diameter (suggesting liquid particles).

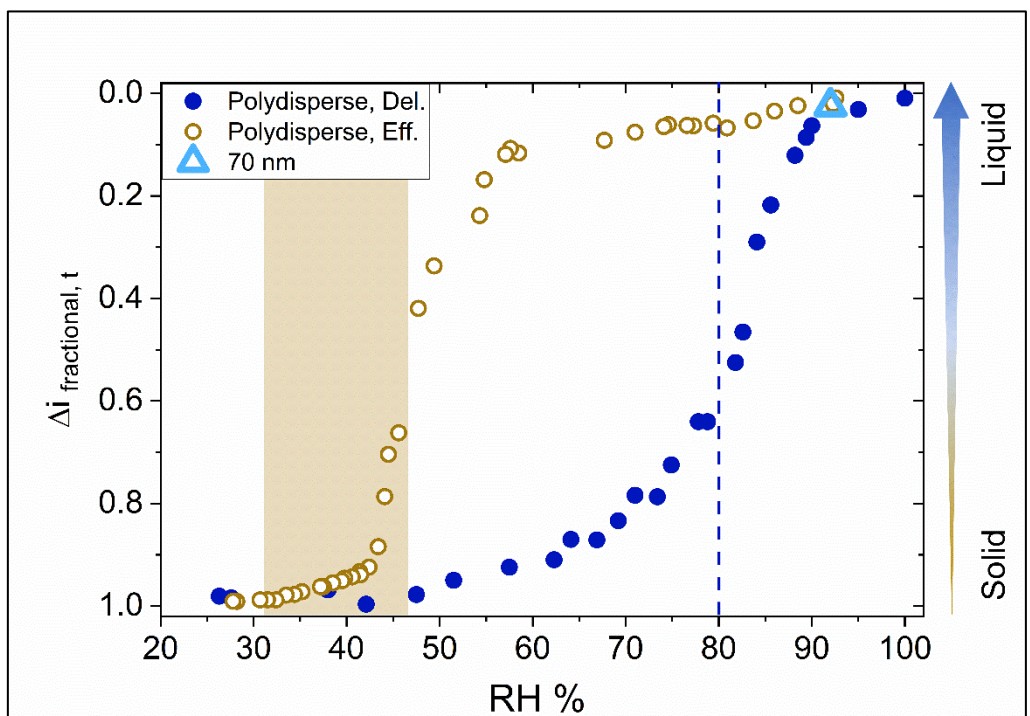

**Figure 1.** Efflorescence relative humidity (o) and deliquescence relative humidity (•) curves for polydisperse (number GMD

= 90 nm, initial dry) ammonium sulfate (AS) aerosol. Shaded area and dashed line indicate literature values for efflorescence and deliquescence, respectively. Particle bounce for monodisperse 70 nm (initial dry diameter) AS ($\triangle$) at 95% chamber RH also shown.



The measured ERH and DRH values for the AS particles were 43% and 87% chamber RH respectively, in good agreement
with reported values of 80 - 82% RH for DRH, and 31 - 48% RH for ERH (Ciobanu et al., 2010; Gao et al., 2006; Brooks et
al., 2002). The decreased impactor RHs (relative to the chamber RH) played a negligible role here given the reported GMD of
this AS particle size distribution and the relatively low ERH of AS. As a note, during experiments the chamber RH was
increased or decreased at a rate of 1% per min and it was not confirmed if equilibrium was established in the chamber after
each RH increment, and the exact crystalline or amorphous character of the specific AS aerosol used here was not determined.
Our ERH and DRH values are slightly higher compared to published literature values likely due to the relatively higher rate
of RH increase (during deliquescence) and decrease (during efflorescence), which has been shown to shift measured glass
transition relative humidity values for glassy solids (Mikhailov et al., 2009). Nevertheless, full and clear deliquescence and
efflorescence of the AS aerosol was observed utilizing this modified method.

When using monodisperse AS particles of 70 nm (initial dry diameter prior to injection into humid chamber), particle bounce
was also shut down at a chamber RH >90%. To verify which ELPI impaction stages were impacted by the monodisperse AS
particles, currents generated over the course of six minutes by each impaction stage were examined. Average currents from
each stage were normalized to allow for easy comparison (Fig. 2). Due to the inherently lower concentrations of monodisperse
AS aerosol, we were not able to measure a DRH curve before particle numbers within the chamber fell (due to dilution) below
ELPI measurement limits. Interestingly, stage 5 ($D_{50}$ = 94.1 nm) experienced the highest current values, likely due to significant
growth of the AS particles in the humid chamber, followed by stages 6 and 4. The maximum impactor stage RH for stages 4
and 5 are 22% and 38%, respectively (Table 2). While the stage 5 RH is nominally at or below the ERH, stage 4 RH is well
below this, yet particle bounce was still shut down ($\Delta i_{\text{fractional, t}}$ <0.05). This suggests the residence time within the lower 6
stages, where large RH drops occur, is not enough to allow for AS aerosol efflorescence. Therefore, AS aerosol of 70 nm
(initial dry diameter) and greater yield ELPI results consistent with AS water uptake and deliquescence, suggesting the validity
of this modified approach to measure particle bounce down to at least 70 nm (initial dry diameter).





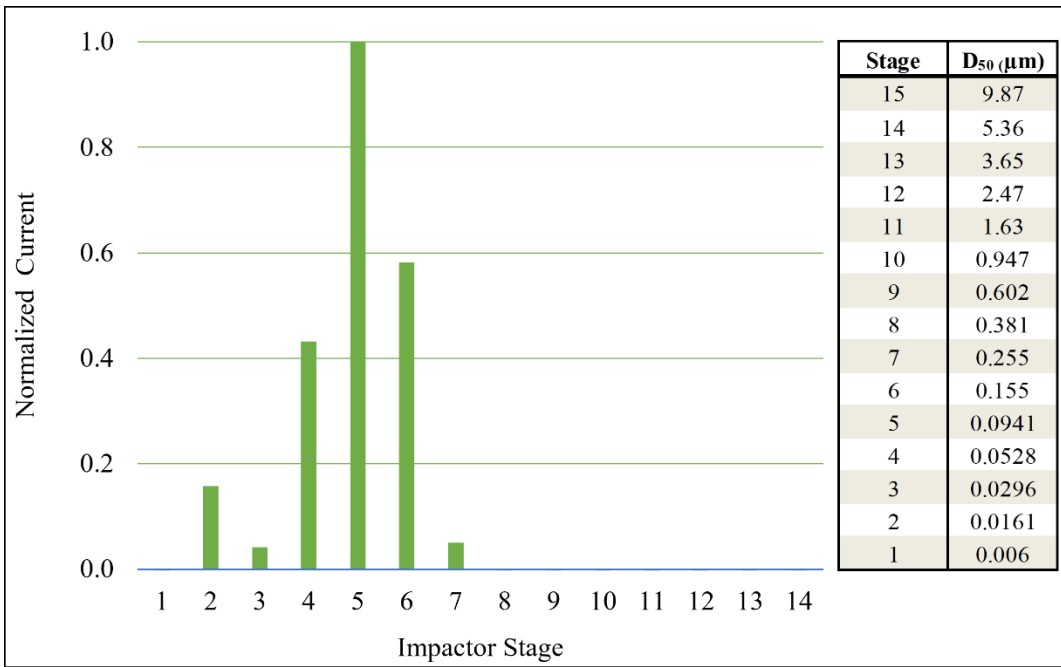

**Figure 2.** Normalized ELPI current values observed for each impactor stage for 70 nm AS particles (initial dry diameter)

sampled from 92% chamber RH.

### 3.2 Sucrose Aerosol

While the particle bounce of AS aerosol behaved as expected as a function of RH, the same was not observed for sucrose

aerosol. For sucrose, a DRH of approximately 85% is well established and measurable water uptake starting at 30 – 40 % RH

has been observed (Robinson et al., 2014), but the ERH has proven difficult to determine (Zobrist et al., 2011; Samain et al.,

2017). Unlike AS, $\Delta i_{\text{fractional, t}}$ for polydisperse sucrose particles (initial dry number GMD = 65 nm) did not approach zero past

the DRH (Fig. 3), which is in direct contrast to other investigators who observed an elimination of sucrose aerosol bounce at

high RH using different methods (Tang et al., 2019; Bateman et al., 2015). However, these methods involved monodisperse

(190 nm and 240 nm) sucrose aerosol and single impactors (as opposed to the ELPI with its cascade impactor setup) with no

impactor pressure drop and hence no RH decrease. As a result, chamber RH and impactor RH remained the same.



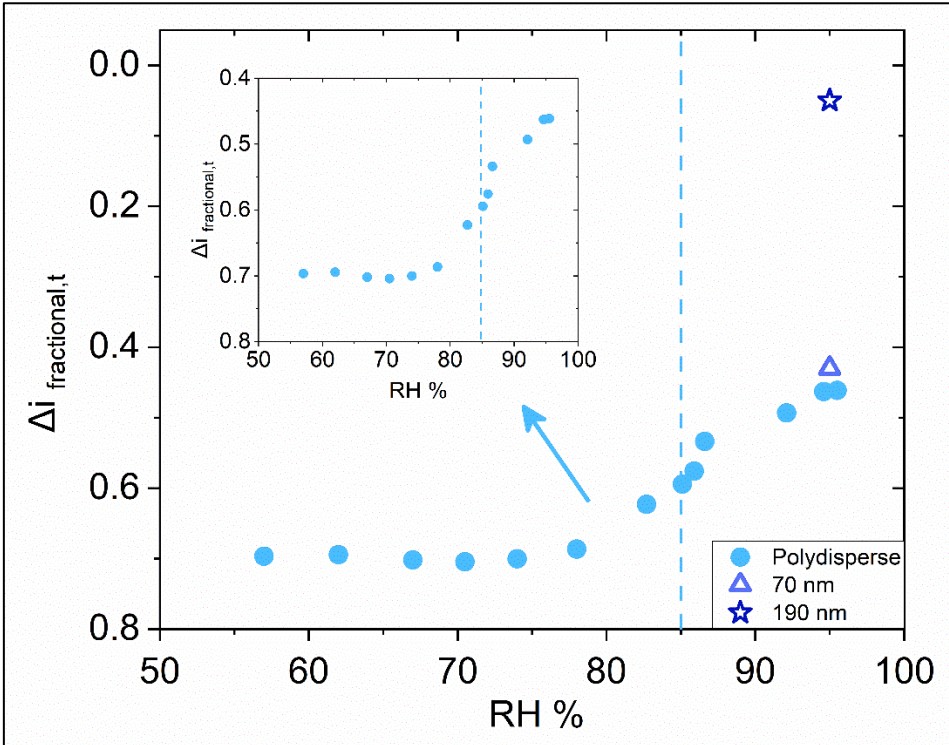

**Figure 3.** Deliquescence relative humidity curve for polydisperse (initial dry number GMD = 65 nm) sucrose (•). Dashed line indicates literature value for deliquescence. Inset for polydisperse sucrose shown for clarity. 70 nm (△) and 190 nm (☆) (both initial dry diameter) at 95% chamber RH also shown.

Monodisperse sucrose aerosol (70 nm and 190 nm, initial dry diameter) was generated and subjected to 95% RH (Fig. 3), well beyond the reported DRH. Due to the inherently lower concentrations of monodisperse sucrose aerosol, a DRH curve was not possible. This monodisperse investigation revealed that $\Delta i_{\text{fractional, t}}$ for 190 nm sucrose particles was very close to zero, indicating that particle bounce was almost entirely shut down at chamber RH > DRH, in contrast to polydisperse (initial dry number GMD = 65 nm) and 70 nm (initial dry diameter) sucrose aerosol. To verify which ELPI impaction stages were impacted by sucrose particles for both the 70 nm and 190 nm experiments, currents generated over the course of six minutes by each impaction stage were examined. Average currents from each stage were normalized to allow for easy comparison (Fig. 4 - 5). For the 70 nm (initial dry diameter) sucrose particles (Fig. 4), stage 1 (filter stage) recorded the highest current, followed by stages 5 and 6. The high stage 1 current is indicative of particle bounce, indicating sucrose particles from higher impaction stages proceeded to bounce following contact with their intended impaction stage. Examining the impaction stage RH for stages 5 and 6 (Table 2) shows a maximum of 68% RH for stage 6, and a maximum of 38% RH for stage 5, assuming 100% chamber RH. Thus, it stands to reason that 70 nm (initial dry diameter) sucrose particles took on water in the humid (95% RH)





chamber, then proceeded to travel through the ELPI and were able to dry sufficiently in the vicinity of stage 5 to subsequently
engage in particle bounce. Hence, $\Delta i_{\text{fractional, t}}$ was observed to remain above zero (0.45) even at high chamber RH.

On the other hand, ELPI current data for the 190 nm (initial dry diameter) sucrose aerosol (Fig. 5) indicates minimal current
recorded on stage 1, with most current recorded on stages 7 and 8. These stages have a maximum RH of 88% and 96%,
respectively, which is above the DRH. Therefore, the larger 190 nm (initial dry diameter) particles, which have grown in the
high chamber RH environment, impact on stages with higher RH closer to that of the chamber and upstream RH. At these
higher impaction stage RHs, the sucrose aerosol did not dry prior to impaction and a $\Delta i_{\text{fractional, t}}$ close to zero was observed
(0.05) at high chamber RH.

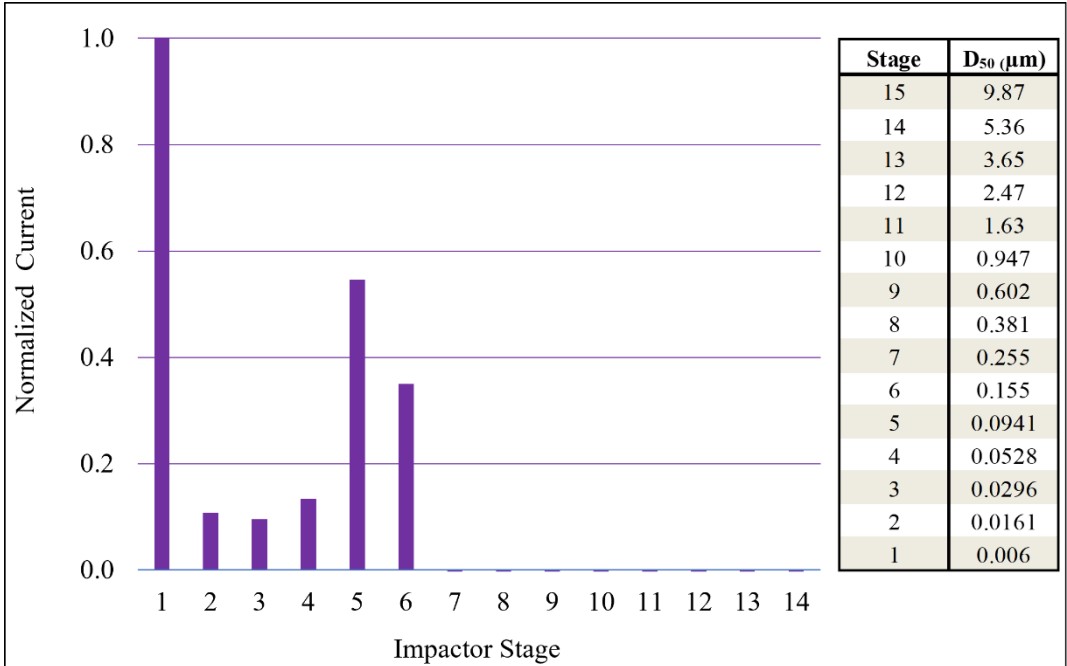

| Stage | $D_{50}$ (μm) |
|---|---|
| 15 | 9.87 |
| 14 | 5.36 |
| 13 | 3.65 |
| 12 | 2.47 |
| 11 | 1.63 |
| 10 | 0.947 |
| 9 | 0.602 |
| 8 | 0.381 |
| 7 | 0.255 |
| 6 | 0.155 |
| 5 | 0.0941 |
| 4 | 0.0528 |
| 3 | 0.0296 |
| 2 | 0.0161 |
| 1 | 0.006 |

**Figure 4.** Normalized ELPI current values observed for each impaction stage for 70 nm (initial dry diameter) sucrose particles
sampled from 95% chamber RH. High stage 1 current indicative of particle bounce.





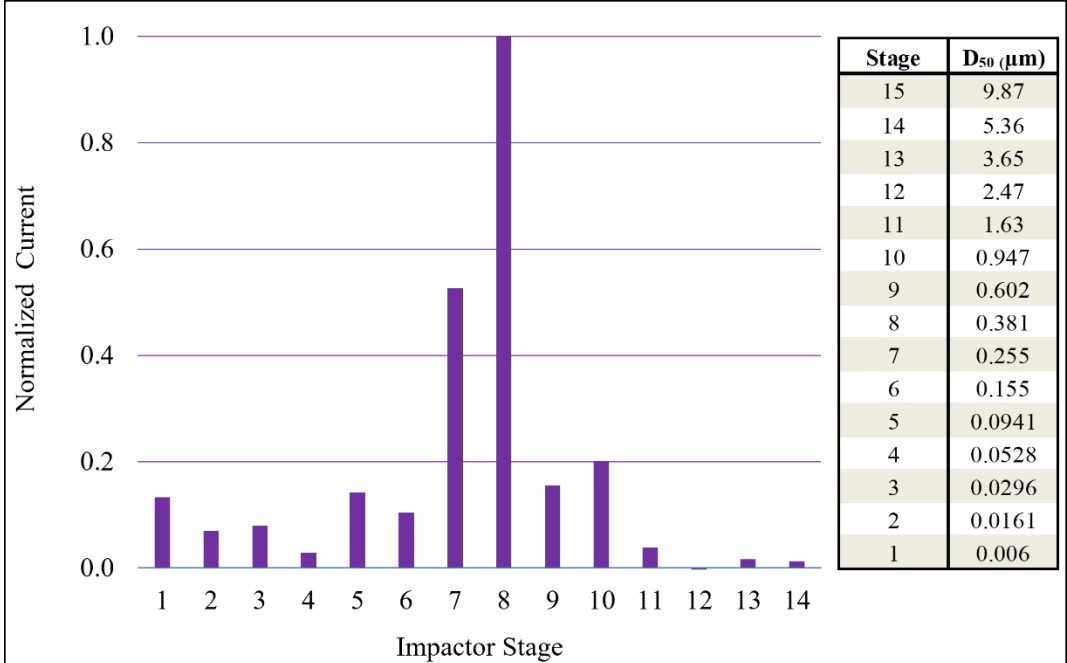

**Figure 5.** Normalized ELPI current values observed for each impaction stage for 190 nm (initial dry diameter) sucrose particles
sampled from 95% chamber RH.

### 3.3 α-pinene Derived Secondary Organic Aerosol

Similar observations were made for polydisperse α-pinene derived SOA at $C_{SOA}$ ranging from 5 to 2,420 µg m$^{-3}$ (Fig. 6). Here,
$\Delta i_{fractional, t}$ never reached 0 under any conditions studied, even at chamber RH close to 100%. SMPS measurements indicate
that the dry particle size distribution stayed well above the $D_{50}$ diameter (6.0 nm) of the filter stage (stage 1), indicating that
particles were reaching the filter stage only as a result of particle bounce, and not due to their aerodynamic diameter. Partial
drying of the SOA due to decreased impactor RH under lower pressure likely caused an incomplete shutdown of α-pinene
derived SOA bounce. Incomplete shutdown of α-pinene derived SOA bounce at high RH is in accord with other investigators
who utilized an impactor setup that featured a drop in RH due to low pressures (Kidd et al., 2014; Saukko et al., 2012b; Saukko
et al., 2015; Slade et al., 2019). Reports that focused on either mobility selected aerosol (monodisperse SOA) (Pajunoja et al.,
2015; Bateman et al., 2014) or on single, relatively large $D_{50}$ impactors without a pressure drop show that α-pinene derived
SOA ceases to bounce at high RH. For example, Bateman et al. (2015) reported on polydisperse measurements, though these
were limited to particle sizes greater than the $D_{50}$ of their β-impactor (~278 nm).





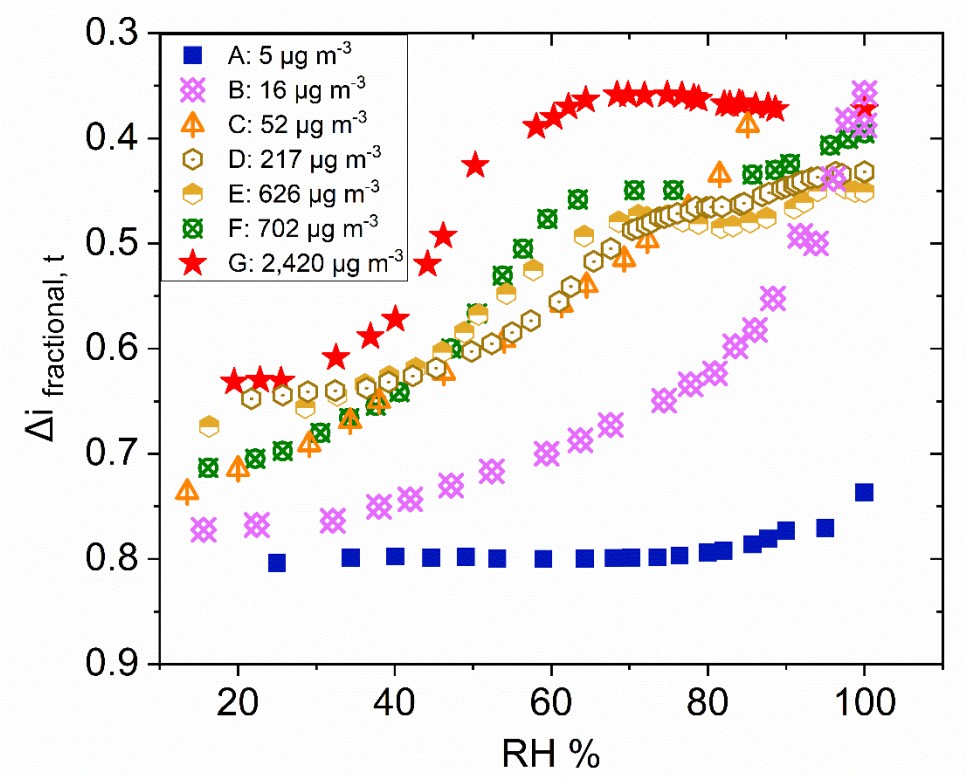

**Figure 6.** $\Delta i_{fractional,\,t}$ for α-pinene derived SOA at various $C_{SOA}$, from 5 µg m⁻³ to 2,420 µg m⁻³, as a function of chamber RH.

SOA derived from α-pinene and other VOC precursors do not exhibit clear deliquescence and efflorescence steps or hysteresis due to their amorphous character (Saukko et al., 2015). Still, further investigating the approximate inflection point RH of each

α-pinene derived SOA experimental run (Fig. 6) and graphing it as a function of the respective $C_{SOA}$ and GMD values yields a clear dependence of both $C_{SOA}$ and GMD on the $\Delta i_{fractional,\,t}$ inflection point RH (Fig. 7; same color scheme used as Fig. 6). The approximate inflection point RH serves as an indicator as to when a significant change in relative particle bounce was observed; though, as previously discussed, particle bounce was never shut down for any α-pinene derived SOA runs.






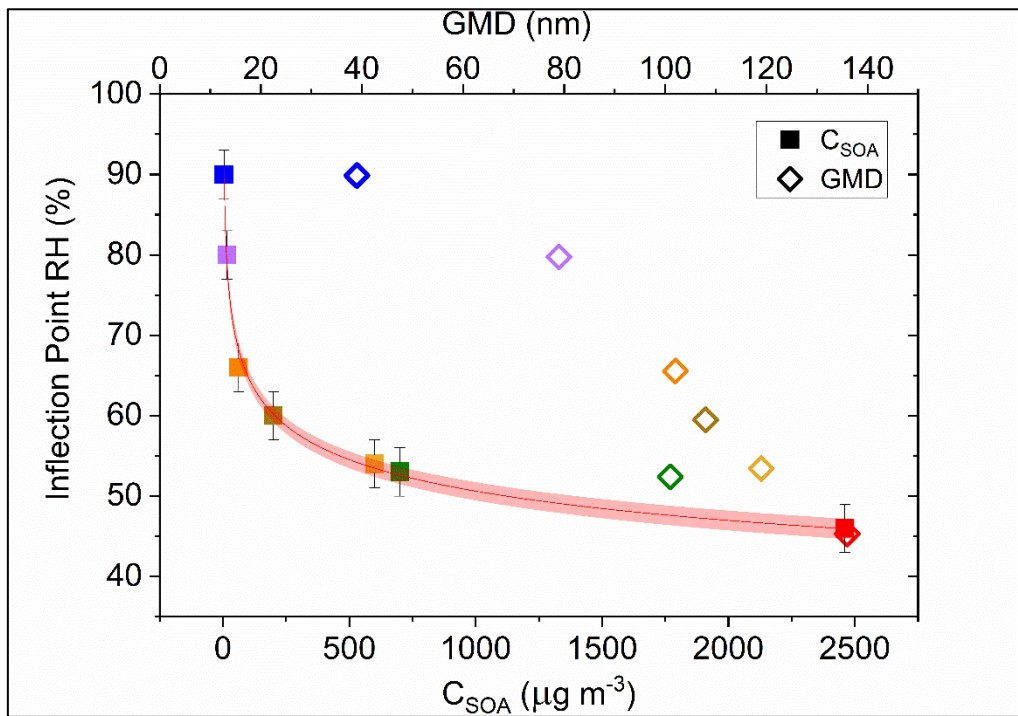

**Figure 7.** Δi fractional, t inflection point RH (%) as a function of SOA mass loading (C_SOA; μg m⁻³) (■) and number geometric mean diameter (GMD; dry diameter; nm) (◊) of SOA distribution. Both C_SOA and GMD observed immediately prior to start of humidification. Error bars represent ± 3% and red band indicates 95% confidence band. Color scheme matches Fig. 6.

The apparent $C_{SOA}$ and GMD trends (Fig. 7) can be explained by examining ELPI current values for all experimental runs (A – G) at the approximate Δi fractional, t inflection point RH (Fig. 8; same color scheme as Fig .6). Here, the current values of each impactor stage at the approximate inflection point RH are shown for each experimental run. Due to persistence of particle bounce, stage 1 (filter stage) exhibited the highest relative current. Ignoring stage 1 and examining the stage with the next highest relative current is indicative of which stage that α-pinene derived SOA are initially impacting on (which is dependent

on $C_{SOA}$ and GMD). This initial impaction stage also corresponds to a specific aerodynamic $D_{50}$ diameter and impactor RH (Table 1). With the exception of A (which maintained a relatively constant Δi fractional, t even at <90% RH), the inflection point RH (which references chamber RH) for each experimental run corresponded to an impactor RH of 37 % - 44%. Therefore, the inflection point RH and the associated drop in Δi fractional, t observed for each experimental run occurred when the initial impactor stage RH reached 37 % - 44 %, regardless of $C_{SOA}$ and chamber RH. As chamber RH continued to increase,

Δi fractional, t decreased until it became constant (at 0.35 – 0.45), at which point the relevant initial impactor stage RH began to reach its maximum RH (Table 1). With an impactor stage RH significantly below the chamber RH, SOA was likely able to dry, leading to a persistence of particle bounce (represented by the Δi fractional, t of 0.35 – 0.45). Experiment A, with its low

C$_{SOA}$ and an initial dry GMD of 39 nm, consisted of particles impacting on stage 4, which has a maximum impactor RH of 22%. As a result, drying was likely more pronounced, and Δi $_{fractional, t}$ remained much higher compared to the other

experimental runs. As a consequence, reliable particle bounce measurements for α-pinene derived SOA at atmospherically relevant C$_{SOA}$ (5 µg m$^{-3}$ initial dry C$_{SOA}$) and size distributions (GMD = 39 nm initial dry) and greater, sampled from high chamber RH, are not possible with this ELPI setup.

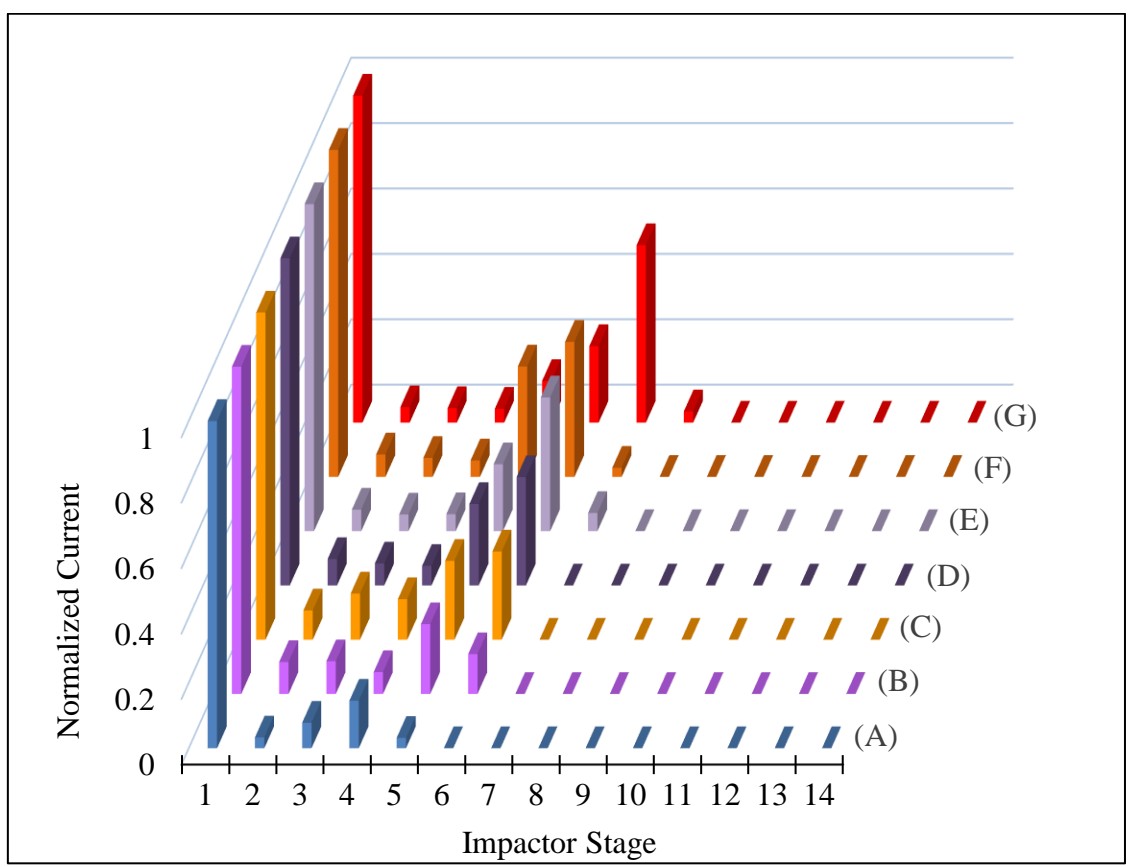

**Fig .8** Normalized ELPI current values observed for each impactor stage, for each α-pinene derived SOA experiment (A-G) at the approximate Δi $_{fractional, t}$ inflection point RH. Stage 1 remained highest due to persistence of particle bounce. Color scheme matches Fig .6.

## 3.4 Aerosol Kelvin Effect

Measurements presented here also include particle sizes where the Kelvin effect (increased equilibrium vapor pressure above a curved surface relative to a flat surface) may be significant in delaying water condensation onto the smaller particles. The Kelvin effect increases in importance as particle diameter decreases (Saxena et al., 1986), especially below approximately 50



nm (Park and Lee, 2000). This may be of special importance for low $C_{SOA}$ (approximately 5 µg m$^{-3}$ based on α-pinene derived SOA results herein) situations, where the dry number GMD was approximately 39 nm. This implies a large share of particles

at diameters expected to be significantly impacted by the Kelvin effect. A more appropriate review therefore could be made for the higher $C_{SOA}$, where GMD was above 150 nm. However, even for the case of the highest $C_{SOA}$ studied here (2,420 µg m$^{-3}$), there was no appreciable particle concentration (< 2% of total particle number density) below a diameter of 50 nm. Therefore, the persistence of bounce cannot be attributed to the Kelvin effect. Furthermore, no correlation between minimum $\Delta i_{fractional, t}$ achieved and percentage of the number concentration of particles below 100 nm was observed (Fig. 9). This is in

agreement with the results of Pajunoja et al. (2015) who showed little difference in water uptake behavior between particles of 50 and 100 nm.

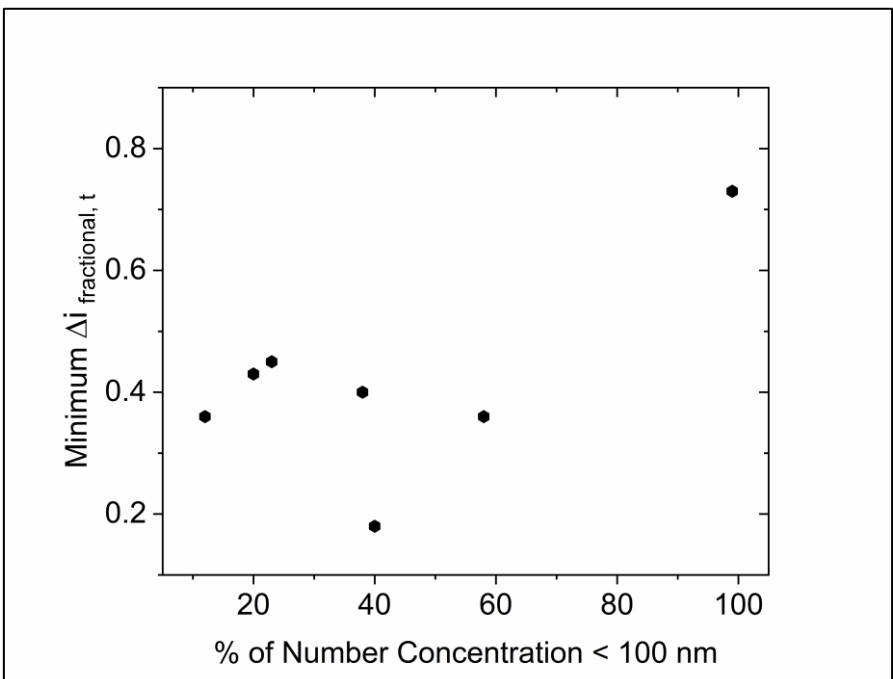

**Figure 9.** Minimum $\Delta i_{fractional, t}$ as a function of percentage of number concentration of particles (within a distribution) below

100 nm in aerodynamic diameter.

### 3.5 Aerosol Drying

A fully deliquesced particle can be viewed as a solute in an aqueous solution, and this solution can experience evaporation and allow for nucleation of crystals or the formation of amorphous particles (Gregson et al., 2019b). This is a complex phenomenon

important to not just the atmospheric science community, but also for the pharmaceutical and food product industries, and is





difficult to probe in situ (Fu et al., 2012). Varying aerosol drying kinetics, which are dependent on aerosol composition (Archer et al., 2020), likely contributed to the particle bounce results discussed here. Gregson et al. (2020) recently investigated the water evaporation rates and crystallization for sodium chloride and sodium nitrate aerosol droplets and were able to observe nucleation and crystallization while also studying how drying temperature, RH, and solute concentration affect nucleation

time. However, a minimum of 25 µm particle diameter was required, well beyond both atmospheric relevance and ELPI (and other impactors) capabilities. Other recent work by Gregson et al. (2019a) involved studying competing evaporation rates of multiple component aerosol droplets (minimum 25 µm) in water, ethanol mixtures. Archer et al. (2020) recently probed the drying kinetics and particle formation for silica aerosol droplets and found that gas phase drying conditions, temperature, and RH impact solvent evaporation rate. A minimum of 25 µm particle diameter was required as well. Robinson et al. (2020)

recently studied drying kinetics and nucleation in evaporating sodium chloride and sodium nitrate aerosols and concluded that sodium chloride recrystallizes upon drying, but not sodium nitrate, indicating differences in the competition between drying rate and nucleation kinetics between the two systems. Once again, minimum droplet sizes of 20 – 25 µm were required. Therefore, while novel advancements have been made recently, there remains the need to carry out experimental in situ studies probing drying, evaporation, crystallization, and nucleation processes for sub-µm organic and inorganic particles of

atmospheric relevance. At this time, further work is warranted to fully understand the different water uptake and loss mechanisms and rates experienced by sub-µm AS aerosol, sucrose aerosol, and α-pinene derived SOA and how these may help to better explain the results reported herein.

## 4 Conclusions and Further Remarks

We demonstrate that technical specifications inherent in the current ELPI design, which relies on a set of cascade impactors

with both decreasing downstream pressure and impaction stage RH, can prevent accurate particle bounce measurements for aerosol sampled from a high RH (> 90%) environment. Specifically, sucrose aerosol below 190 nm (initial dry diameter) and α-pinene derived SOA populations of atmospheric relevant $C_{SOA}$ (5 µg m$^{-3}$ initial dry $C_{SOA}$) and size distributions (number GMD = 39 nm initial dry) and greater exhibited persistent particle bounce when sampled from a RH > 90% Teflon reaction chamber. For AS aerosol, this limit was 70 nm (initial dry diameter). More generally, for sucrose aerosol and α-pinene derived

SOA, reliable particle bounce measurements cannot be made with the ELPI setup if the ERH or equivalent drying RH is above the initial impaction stage RH. The residence times within the lower impaction stages, where RH decreases, likely allowed for sufficient drying to occur to allow particle bounce to persist. For AS aerosol, this residence time is likely too short to allow for sufficient drying, which may be due to differing drying, evaporation, crystallization (if applicable), and nucleation processes. A more in-depth understanding of these processes at the sub-µm scale for individual aerosol systems would aid in better

understanding the reasons for the differing particle bounce behaviors observed for AS aerosol, sucrose aerosol, and α-pinene derived SOA and permit developing corrections in order to be able to apply this revised bounce method. Thus, additional experimental work is needed to determine the viability and improve the accuracy of indirect measurements of water uptake of other atmospherically relevant aerosol populations.



*Author Contributions.* Kevin Fischer and Giuseppe Petrucci designed experiments. Kevin Fischer carried out experiments and wrote manuscript. Giuseppe Petrucci contributed with editing and reviewing manuscript.

*Competing Interests.* The authors declare that they have no conflict of interest.

*Acknowledgements.* The authors gratefully acknowledge the support of the National Science Foundation under Grant CHE-1709751

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
