# Peer review of "Utilizing an Electrical Low Pressure Impactor to Indirectly Probe Water Uptake via Particle Bounce Measurements"

_Atmospheric Measurement Techniques, 2021_

## Author Comment (AC1)

Utilizing an Electrical Low Pressure Impactor to Indirectly Probe Water Uptake via Particle Bounce Measurements Fischer, Petrucci

**Response to RC2: 'Comment on amt-2021-35', Anonymous Referee #3, 02 Jun 2021:**

We thank anonymous referee #3 for taking their time to carefully review our manuscript and provide detailed feedback.

"The authors examine the use of an electrical low pressure impactor (ELPI) to study phase changes in aerosol, focusing on efflorescence, deliquescence, and the formation of viscous amorphous liquid phases. They demonstrate results for ammonium sulfate that are consistent with expectations based on hygroscopic growth and phase change RH's. For sucrose, they show that the starting size is important due to the fact that the RH in the impactor decreases from the inlet to the lowest stage. This results in smaller particles showing more bounce, indicating the formation of a solid or semi-solid phase. Given the RH changes in the impactor, it seems that the observations can be explained in terms of RH-dependent size and viscosity for all these systems, along with size itself for the SOA system."

We would also like to emphasize the  $D_{50}$  diameters for each stage, which decrease as particles move down stream within the ELPI.

"Overall, the observations appear to make sense based on the described experiments. However, unless the RH can be made uniform across the impactor stages, it is not clear to me that this method could be reliably used for exploring RH-dependent phase. The authors do a good job at stressing the problems relating to the variable RH, but who is this work intended for? As far as I can tell, the only researchers using the ELPI for viscosity and phase measurements are the authors of this manuscript (please correct me if I'm wrong!)."

Due to the low pressure environment of the lower impaction stages, RH drops are inherent. The atmospheric science community is continuously developing and refining techniques to probe aerosol phase, and this method (or a similar method involving a low pressure impactor setup) has been used by (Järvinen et al., 2014), (Kidd et al., 2014), (Saukko et al., 2012; Saukko et al., 2015), (Virtanen et al., 2010), (Jain and Petrucci, 2015), (Slade et al., 2019), to name a few. One purpose of this manuscript is to highlight potential RH artefacts, which can appear when conducting experiments under high RH conditions.

Utilizing an Electrical Low Pressure Impactor to Indirectly Probe Water Uptake via Particle Bounce Measurements Fischer, Petrucci

"Generally, I think this manuscript should be published with some revision, but the authors need to be more clear about the aims of the work, and describe the broader implications. Are other impactor measurments of phase affected in similar ways? Can the problematic aspects of this experiment be harnessed for new insights?"

Other impactor measurements utilizing impactors that do not feature a pressure drop do not appear to be affected in similar ways, as discussed in the manuscript with references to (Tang et al., 2019) and (Bateman et al., 2015). Though, as noted, these methods have different limitations (such as the requirement for monodisperse aerosol populations only.)

"DRH of 87% is quite a lot higher than the literature - this is not an insignificant difference. Can the authors explain this difference in terms of uncertainty in RH probes or some other factors? 1% RH change per minute seems very fast for such a large chamber - likely this is the source. Where was the RH measured? Why not vary the RH more slowly to better equilibrate the chamber?"

We agree that the RH change per minute is relatively fast and likely accounts for the higher DRH, and this is noted in the manuscript. The RH was measured with a RH probe located inside the chamber (HMT 130, Vaisala Corp., Helsinki, Finland), as noted in the manuscript. Ideally, the RH change can be adjusted more slowly. However, our humidification and drying setup does not allow for this. Furthermore, wall losses within the chamber cause a decrease in aerosol population, meaning there is a limited time to probe our aerosol population.

"The impact design and change in RH means that small particles may enter viscous states before reaching the filter. A small aqueous particle that should splat on stage 4, for example, may instead become viscous and bounce down to stage 1. Authors should try to estimate the amount of expected water loss over 50ms in each stage to estimate impact of this RH gradient. For very small particles, 50 ms can be plenty long enough for gas-phase diffusion controlled evaporation to occur."

We agree that gas phase diffusion controlled evaporation may be occurring. Estimating the expected water loss over 50 ms in each impactor stage would involve many assumptions and approximations. We do not think that this estimation would add any information of value, as we are not attempting to quantify the water loss in this manuscript.

Utilizing an Electrical Low Pressure Impactor to Indirectly Probe Water Uptake via Particle Bounce Measurements Fischer, Petrucci

"In the discussion of the Kelvin effect, providing some estimates of the water activity in the smaller particles as a function of RH would be insightful to see the potential magnitude of this effect."

The purpose of this section is to highlight that the persistence of particle bounce is NOT likely due to the Kelvin effect. Therefore, we do not feel it is appropriate to estimate the water activity of the smaller particles as a function of RH in this work.

**References**:**

Bateman, A. P., Bertram, A. K., and Martin, S. T.: Hygroscopic Influence on the Semisolid-to-Liquid Transition of Secondary Organic Materials, The Journal of Physical Chemistry A, 119, 4386-4395, 10.1021/jp508521c, 2015.

Jain, S. and Petrucci, G. A.: A New Method to Measure Aerosol Particle Bounce Using a Cascade Electrical Low Pressure Impactor, Aerosol Science and Technology, 49, 390-399, 10.1080/02786826.2015.1036393, 2015.

Järvinen, A., Aitomaa, M., Rostedt, A., Keskinen, J., and Yli-Ojanperä, J.: Calibration of the new electrical low pressure impactor (ELPI+), Journal of Aerosol Science, 69, 150-159, <a href="https://doi.org/10.1016/j.jaerosci.2013.12.006">https://doi.org/10.1016/j.jaerosci.2013.12.006</a>, 2014.

Kidd, C., Perraud, V., Wingen, L. M., and Finlayson-Pitts, B. J.: Integrating phase and composition of secondary organic aerosol from the ozonolysis of  $\alpha$ -pinene, 111, 7552-7557, 10.1073/pnas.1322558111 %J Proceedings of the National Academy of Sciences, 2014.

Saukko, E., Zorn, S., Kuwata, M., Keskinen, J., and Virtanen, A.: Phase State and Deliquescence Hysteresis of Ammonium-Sulfate-Seeded Secondary Organic Aerosol, Aerosol Science and Technology, 49, 531-537, 10.1080/02786826.2015.1050085, 2015.

Saukko, E., Lambe, A. T., Massoli, P., Koop, T., Wright, J. P., Croasdale, D. R., Pedernera, D. A., Onasch, T. B., Laaksonen, A., Davidovits, P., Worsnop, D. R., and Virtanen, A.: Humidity-dependent phase state of SOA particles from biogenic and anthropogenic precursors, Atmos. Chem. Phys., 12, 7517-7529, 10.5194/acp-12-7517-2012, 2012.

Slade, J. H., Ault, A. P., Bui, A. T., Ditto, J. C., Lei, Z., Bondy, A. L., Olson, N. E., Cook, R. D., Desrochers, S. J., Harvey, R. M., Erickson, M. H., Wallace, H. W., Alvarez, S. L., Flynn, J. H., Boor, B. E., Petrucci, G. A., Gentner, D. R., Griffin, R. J., and Shepson, P. B.: Bouncier Particles at Night: Biogenic Secondary Organic Aerosol Chemistry and Sulfate Drive Diel Variations in the Aerosol Phase in a Mixed Forest, Environmental Science & Technology, 53, 4977-4987, 10.1021/acs.est.8b07319, 2019.

Tang, M., Chan, C. K., Li, Y. J., Su, H., Ma, Q., Wu, Z., Zhang, G., Wang, Z., Ge, M., Hu, M., He, H., and Wang, X.: A review of experimental techniques for aerosol hygroscopicity studies, Atmos. Chem. Phys., 19, 12631-12686, 10.5194/acp-19-12631-2019, 2019.

Virtanen, A., Joutsensaari, J., Koop, T., Kannosto, J., Yli-Pirilä, P., Leskinen, J., Mäkelä, J. M., Holopainen, J. K., Pöschl, U., Kulmala, M., Worsnop, D. R., and Laaksonen, A.: An amorphous solid state of biogenic secondary organic aerosol particles, Nature, 467, 824-827, 10.1038/nature09455, 2010.

---

## Author Response (AR1)

**Response to Referee 1:**

We thank referee #1 for taking their time to carefully review our manuscript and provide detailed feedback. We have taken into consideration all comments and suggestions and have made appropriate changes where applicable.

**1) clarify whether (and how if any) the defined delta_i_fractional,t can be translated to bounce factor, e.g., through a relationship shown in Figure S1?**

The $\Delta i_{fractional, t}$ cannot be quantitatively translated to bounce factor, as it is a more qualitative representation of particle bounce. It is possible to compare it to the bounce factor method, though, which is shown in Fig S1. We have added text clarifying that a comparison can be found in Fig. S1.

**2) use consistent y axis scales (i.e., 0 to 1 or 1 to 0) for Figures 1 and 6.**

We have revised Figure 6 and the y-axis is now consistent with the y-axis scales found in the previous figures.

**Response to Referee 2:**

We thank referee #2 for taking their time to carefully review our manuscript and provide detailed feedback. We have taken into consideration all comments and suggestions and have made appropriate changes where applicable.

**L184: the author should list the range of DRH/ERH from the literature.**

We agree and have added DRH (80 – 82% RH) and ERH (31 – 48% RH) to the text.

**Fig.2: Can the author discuss a bit more why the current for stage 3 is smaller than stage 2? The trend is a bit weird.**

We agree that this appears a bit peculiar, but is likely due to higher noise on this particular channel relative to the others. The main objective of Fig. 2 is to demonstrate that, even though a majority of ammonium sulfate particles impacted on impactors with impactor RH close to or below the ERH, particle bounce was shut down and therefore no significant drying occurred (in contrast to 70 nm sucrose).

**Once the particle hit the electrometer, can charge transfer happen even if the particle bounce? Will this lead to any error in the calculation of the ELPI current and bounce calculation? For now I believe the author thinks the current from ELPI indicated the particle hit onto the impactor, but it is also possible the particle also bounced back and did not stay on the impactor**

Charge transfer can happen even if the particle bounces. When particles bounce, our method assumes they bounce all the way down to the filter stage (stage 1). When particle bounce occurs, a large amount of current is recorded on the filter stage (stage 1). In fact, as the smooth impaction stages were used here, particle bounce was enabled for the more solid-like particles, while liquid particles do not bounce. This is all discussed more in detail by (Jain and Petrucci, 2015), which is cited in the manuscript.

**Can one particle bounce multitimes on different impactor stages and will this lead to any issue in interpreting the data? The author should clarify this in the manuscript.**

Please see above.

**Figure 6: The y axis is plotted inversely compared with previous figures, which is confusing when comparing other figures. The y axis should be the same descending order as other figures. Please revise.**

We have revised Figure 6 and the y-axis is now consistent with the y-axis scales found in the previous figures.

**L275: did the author compare the inflection RH when the change of bounce occurs with the transition RH reported from other a-pinene SOA bounce studies? If so please list the results of the literature comparison.**

This comparison was not made because other studies were able to observe a full shutdown of particle bounce with their respective impactor setups. For the ELPI here, particle bounce persisted. We do not believe it is appropriate to make this comparison to other impactor setups that do not feature a drop in pressure and RH within the impactors.

**To verify the conclusion that drying RH within the impactor is the reason for different bounce behaviors of a-pinene SOA with different concentrations, I would suggest the author perform a calibration study using sucrose of different sizes. In section 3.2, the author only used monodisperse AS only in one size and monodisperse sucrose only in two sizes. So adding more data points will help strengthen the explanation why the bounce behaviors of a-pinene SOA are different**

While we agree that more data points would certainly assist in our investigation, the choice of 70 nm and 190 nm sucrose was made based off of the $D_{50}$ diameters of the ELPI and the corresponding impactor RH values. The purpose of selecting these two diameters was to choose particles on more extreme ends of the size spectrum. In other words, particles that would impact on impactors having a local impactor RH well below that of the chamber RH, and on the other hand to choose particles that would impact on impactors having a local impactor RH negligibly different from that of the chamber RH. Additionally, sucrose is known to deliquescence and ceases to bounce past its DRH (Bateman et al., 2015).

**Did the author compare this result with that from Grayson et al. Grayson et al. Seem to suggest that the concentration of a-pinene SOA also affect the viscosity while in this study the author did not seem to observe this effect. Any explanations?**

We agree that concentration of a-pinene SOA affects viscosity. However, this was not the focus of this study. Here, physical uptake of water was the focus. Due to the RH artefact experienced by the ELPI, the particle bounce for a-pinene derived SOA sampled from high RH environments cannot be measured accurately, regardless of the concentration. Since particle bounce cannot be measured accurately for a-pinene derived SOA with the ELPI, it is not possible to discern whether or not we observed a difference in particle bounce (which is a surrogate for viscosity).

Bateman, A. P., Bertram, A. K., and Martin, S. T.: Hygroscopic Influence on the Semisolid-to-Liquid Transition of Secondary Organic Materials, J. Phys. Chem. A, 119, 4386-4395, 10.1021/jp508521c, 2015.

Jain, S., and Petrucci, G. A.: A New Method to Measure Aerosol Particle Bounce Using a Cascade Electrical Low Pressure Impactor, Aerosol Sci. Technol., 49, 390-399, 10.1080/02786826.2015.1036393, 2015.

---

## Author Response (AR2)

**Response to Reviewer #3:**

We thank Reviewer #3 for taking their time to carefully review our manuscript and provide detailed feedback.

**Line 106. One of the major takeaways of this manuscript is that ELPI particle bounce measurements need to be corrected for variable RH at different impaction stages. The authors should describe the ELPI in greater detail in this section, so the reader can gain a greater understanding of the apparatus, possibly including a diagram in the main text or SI. In particular, the assumption (line 125) that all particles that bounce from any stage end up on the filter stage should be explained in more detail, as it is key to the data acquisition but is not obvious (at least to me). If there is any experimental data to backup this assumption, that should be presented as well.**

We agree, and have provided additional information for the ELPI in the Supplemental text. The reference in the Supplemental text and the cited literature within that reference provide additional information regarding the ELPI operating principle.

**Line 167. Figure 1 seems to show that the deliquescence curve, which should lead to particles exhibiting the same behavior between <40% RH up to the deliquescence point at 80% RH (at which point a discontinuity should occur), actually exhibits an intermediate change in the fractional delta_I between 40 and 80% RH. This is very surprising to me and the authors should offer some physical explanation to why this occurs. Without such an explanation, it is not clear to me how reliable these measurements to probe particle phase state, including when attempting to interpret SOA data later in the manuscript.**

During experiments the chamber RH was increased at a rate of 1% per minute, and a homogenous RH within the chamber could not be guaranteed. ELPI sampling also occurred at roughly 10 Liters per minute. Locally higher concentrations of gaseous water likely allowed for partial deliquescence (when viewing the chamber system as a whole) to occur earlier, prior to reaching 80% RH as measured by the humidity probe in one location of the chamber. Nevertheless, full and clear deliquescence and efflorescence transitions of the AS aerosol were observed utilizing this modified method. Again, the applicability of this method is not intended so much to determine accurately RH or deliquescence/efflorescence of any given aerosol, but rather to be able to infer the general phase state of SOA aerosol during its formation and aging.

**Line 185. I disagree with the statement that the efflorescence and deliquescence RHs are in good agreement with previous reports. While ERHs can vary, DRH should be close to 80%, and 87% is quite high. Some explanation of why this discrepancy might have arisen and how precise the authors believe the RH measurements are in light of this discrepancy should be included.**

This likely occurred due to the relatively higher rate of RH increase (during deliquescence). This has been shown to shift measured glass transition relative humidity values for glassy solids to higher values (Mikhailov et al., 2009). Furthermore, a homogenous RH within the chamber could not be guaranteed as the RH probe only measured one location within the chamber. We have now added a note in the manuscript.

**Line 198. The authors seem to operate their chamber essentially in batch mode where the RH is changed by diluting the particles, which causes an issue with low particle concentrations, as stated here. As a suggestion for future studies, the authors might consider operating their chamber as a continuously mixed flow reactor with particles continuously injected while sampling, and modifying the RH at the inflow of the chamber using a Nafion dryer, which should circumvent this issue.**

We appreciate this suggestion and have begun preliminary work to redesign our experimental setup to convert our chamber into a continuously mixed flow reactor.

**Line 203. Here and at several other points in the manuscript, the authors refer to particle bounce as being "shut down". This terminology seems imprecise and casual to me, and I suggest other terminology such as "eliminated", "nearly eliminated", "greatly reduced", or another term.**

We agree, and have replaced "shut down" with "eliminated".

**Line 231. Here and elsewhere in the manuscript, polydisperse GMD are reported but no measure of the polydispersity is included, which is important for the interpretation. Some measure of the polydispersity, such as a standard deviation, or, even better, actual measured particle distributions, should be included for every polydisperse sample.**

We agree with the reviewer and have edited the manuscript to include typical values of polydispersity (as measured by the geometric standard deviation). Also, typical particle size distributions have been provided in the Supplemental Information (**Figures S4-S6**).

**Section 3.4. The discussion of the Kelvin effect is currently vague and qualitative. I am not sure how much Figure 9 adds to the discussion and would consider removing it. What would be more quantitatively interesting and relevant would be seeing how much the RH changes from the RH a bulk phase or large (>1 micron) particle experiences for each of the particle diameters in the SOA data set. If it is only a few percentage points for the smallest sizes, then this effect should be negligible. However, if it is greater than this, then it would be worthwhile to further correct the ELPI RH for these smaller stages, with this Kelvin effect correction factor as well.**

We included this section to show that even for the case of the highest $C_{SOA}$ studied here (2,420 µg m$^{-3}$), there was no appreciable particle concentration ($< 2\%$ of total particle number density) below a diameter of 50 nm, which one might argue would not be of a critical size necessary for water uptake and so would remain solid and bounce. We therefore reason that the persistence of bounce cannot be attributed to the Kelvin effect. Figure 9 shows that there is no correlation between minimum $\Delta i_{fractional, t}$ achieved and percentage of the number concentration of particles below 100 nm  (Figure 9), further supporting our suggestion that the Kelvin effect is not responsible for the observed residual bounce. As such, we believe that Figure 9 is important to show. Bulk phase or large ($> 1$ micron) particles were not the focus here, but certainly warrant their own studies, as they may be more important in the form of organic-coated salt particles in the marine environment. Their behavior with respect to the ELPI and RH may certainly differ from the behavior of the fine particles studied here.

**Section 3.5. This section partially addresses what I consider to be one of the two main takeaways of this manuscript: that the varying RH across ELPI stages could be used to study the kinetics of water uptake/evaporation in aerosol particles. However, this section mostly talks in general about prior literature results. Any relevant information here would fit better in the introduction and should be moved there. What would strengthen this section, and the manuscript as a whole, would be a discussion of how exactly the ELPI could be used to study water uptake/evaporation. Can the residence time in each stage be varied (by flow rate, or by modifying the apparatus itself) to study how long it takes for a particle to effloresce, for instance? Are there other experiments that would be particularly interesting to run in this area? This information would be of great interest to me.**

Unfortunately, our ELPI instrument is commercially produced and adjustments to the residence time via modifications to the flow rate or the apparatus itself are not possible, as this would alter instrumental parameters and affect the internal calculations. However, the ideas presented in this manuscript open the doors to future studies with custom built impactors that allow for modifications of parameters such as flow rate or particle resident times. This would allow for detailed, future studies of water uptake/evaporation. With this manuscript, we aim to show that the Dekati ELPI may not be suitable for measurements under high ambient RH and offer a word of caution for the community. We have edited the manuscript to include this section in the introduction, per the Reviewer's suggestion.

---

## Author Response (AR3)

Dear Dr. Tang,

than you for your editorial comments. We have responded to each, as described below:

1. Page 1, line 8: change "display" to "displays"?

Correction has been made

2. Page 5: would it be better to move Figure S1 to the main text?

Figure S1 has been moved to text and all figure numbers updated

3. Page 6, line 150: change "an RH drop" to "a RH drop"?

Correction has been made

4. Page 15, line 311: change "Fig. 8" to "Figure 8" for consistence?

Correction has been made

5. The last page in the supporting information does not have a caption for the figure. In addition: it seems to have two panels, but the lower panel is not properly shown. Please check.

Caption has been added.